# Diverse Possibilities of Si-Based Agent, a Unique New Antioxidant

**DOI:** 10.3390/antiox12051061

**Published:** 2023-05-08

**Authors:** Yoshihisa Koyama, Yuki Kobayashi, Hikaru Kobayashi, Shoichi Shimada

**Affiliations:** 1Department of Neuroscience and Cell Biology, Osaka University Graduate School of Medicine, Suita 565-0871, Japan; shimada@anat1.med.osaka-u.ac.jp; 2Addiction Research Unit, Osaka Psychiatric Research Center, Osaka Psychiatric Medical Center, Osaka 541-8567, Japan; 3SANKEN, Osaka University, Suita 567-0047, Japan; koba42@sanken.osaka-u.ac.jp (Y.K.); h.kobayashi@sanken.osaka-u.ac.jp (H.K.)

**Keywords:** silicon, antioxidant, reactive oxygen species, hydrogen, reactive sulfur species, anti-inflammation, anti-apoptosis, anti-fibrotic

## Abstract

Antioxidant therapy is an effective approach for treating diseases in which oxidative stress is involved in the onset of symptoms. This approach aims to rapidly replenish the antioxidant substances in the body when they are depleted due to excess oxidative stress. Importantly, a supplemented antioxidant must specifically eliminate harmful reactive oxygen species (ROS) without reacting with physiologically beneficial ROS, which are important to the body. In this regard, typically used antioxidant therapies can be effective, but may cause adverse effects due to their lack of specificity. We believe that Si-based agents are epoch-making drugs that can overcome these problems associated with current antioxidative therapy. These agents alleviate the symptoms of oxidative-stress-associated diseases by generating large amounts of the antioxidant hydrogen in the body. Moreover, Si-based agents are expected to be highly effective therapeutic drug candidates because they have anti-inflammatory, anti-apoptotic, and antioxidant effects. In this review, we discuss Si-based agents and their potential future applications in antioxidant therapy. There have been several reports of hydrogen generation from silicon nanoparticles, but unfortunately, none have been approved as pharmaceutical agents. Therefore, we believe that our research into medical applications using Si-based agents is a breakthrough in this research field. The knowledge obtained thus far from animal models of pathology may greatly contribute to the improvement of existing treatment methods and the development of new treatment methods. We hope that this review will further revitalize the research field of antioxidants and lead to the commercialization of Si-based agents.

## 1. Introduction

Reactive oxygen species (ROS) produced by respiration and immunoreactions are highly oxidizing. Under normal conditions, they are eliminated by in vivo antioxidant substances (superoxide dismutase, glutathione, and catalase). However, when excessive ROS production is caused by chronic diseases, viral or bacterial infection, and gluttony, the antioxidant mechanism in the body cannot eliminate ROS sufficiently. Consequently, important cellular components are oxidized by ROS, resulting in tissue damage and dysfunction. “Oxidative stress” is a pathological condition caused by a combination of overproduced ROS and depleted antioxidant substances in the body [1,2,3]. Oxidative stress is involved not only in the onset and symptomatic exacerbation of various diseases—such as inflammatory, metabolic, neurological, and ischemic diseases and cancer—but also in aging [1,2,3]. Accordingly, it is useful to administer exogenous antioxidants as replacements for depleted in vivo antioxidant substances [4]. Unfortunately, many antioxidants—such as vitamin C (VC), vitamin E (VE), polyphenols, and carotene—can have adverse effects, because they also remove useful biogenic ROS (superoxide, hydrogen peroxide, etc.) that are involved in normal physiological activities [5,6,7,8,9,10]. For example, overdoses of VC and VE induce nausea, vomiting, and increased cancer risk [11,12,13]. Moreover, when the effective doses of VC and VE, as determined in animal experiments, are converted to their human equivalents, the resulting doses are much higher than the upper tolerable limits for VC (2 g/day) or VE (1 g/day) recommended by the Food and Nutrition Board of the National Academic Medicine [14]. Paradoxically, excess vitamin C acts as a prooxidant and can worsen symptoms [15]. Thus, even if promising results are obtained through basic research, the clinical application of antioxidant therapy presents challenges. Given the above information, the discovery of hydrogen as an antioxidant in 2007 was highly significant [16].

Hydrogen selectively eliminates harmful ROS such as hydroxyl radicals and has known efficacy in alleviating the pathology of various diseases [17,18]. Moreover, with no reported side effects, hydrogen antioxidant therapy is safe [19]. However, the in vivo administration methods can be improved. Currently, there are two main methods: hydrogen gas inhalation and drinking water. The former is limited to specific settings because of the risk of explosion, whereas in the latter, only small amounts of hydrogen can be dissolved in water (1.6 ppm), which cannot sustain elevated hydrogen concentrations over long periods because of its excellent hydrogen permeability. In contrast, our Si-based agent reacted with water to generate hydrogen and made it possible to fill the rodent intestinal tract with a large amount of hydrogen via the oral administration of its agent-containing diet (Figure 1) [20,21]. So far, Si-based agents have alleviated the pathology of ROS-mediated diseases, such as model colitis (UC), Parkinson’s disease, renal failure, and facial paralysis [21,22,23]. We believe that Si-based agents offer effective solutions to the above-mentioned challenges in hydrogen antioxidant therapy.

Previous reports have demonstrated the efficacy of Si-based agents for each disease. However, there have been no reports investigating the detailed action mechanism of the antioxidant effect and its various properties in Si-based agents. In this review, a cross-sectional analysis that transcends the framework of each paper has revealed an interesting hypothesis that Si-based agents have a variety of effects, and that these effects are exerted in various organs via the circulatory system. We describe the characteristics and efficacy of Si-based agents in detail and discuss the potential of SI-based agents in treating ROS-mediated conditions and their hypothetical action mechanisms.

## 2. Development of a New Antioxidant Si-Based Agent

Our Si-based agent was fabricated from polycrystalline Si powder (Osaka Titanium technologies Co., Ltd., Osaka, Japan; Si 5N Powder 2–45 µm). After bead-milling the Si powder in ethanol, a surface treatment was performed using a hydrogen peroxide solution to enhance the surface reactivity. To improve the safety of the Si-based agent, an agglomeration treatment was performed. The average crystallite size was 20–30 nm. Figure 1 shows the scanning electron micrographs of the Si-based agent. It was clearly seen that the Si-based agent consisted of agglomerates of Si nanopowder. The size of the agglomerates was 0.1–20 μm.

Figure 2 shows the volume of hydrogen generated by reaction of the Si-based agent with a pH 8.2 aqueous solution at 36 °C [20,21]. This hydrogen generation reaction continued for more than 24 h.

It has been observed that the hydrogen generation rate significantly increases with the pH of the reacting solutions, while the pH of the reacting solutions themselves remains unchanged [20]. From these experimental results, we derived the following reaction formulas for hydrogen generation:Si + 2OH^−^ → SiO_2_ + 2H (or H_2_) + 2e(1)
2H_2_O + 2e → 2OH^−^ + 2H (or H_2_)(2)
Si + 2H_2_O → SiO_2_ + 4H (or 2H_2_)(3)

Because the rate of Reaction (1) is much lower than that of Reaction (2), the overall hydrogen generation reaction (Reaction (3)) rate is determined by Reaction (1): In Reaction (1), OH^−^ ions react with Si, and therefore, the reaction rate greatly increases with an increase in the concentration of OH^−^ ions, i.e., pH. Additionally, in Reaction (1), OH^-^ ions are consumed, while in Reaction (2), they are generated in an equivalent quantity, and hence, after the overall Reaction (3), the pH of the resulting solution is unchanged.

Figure 3 shows the mechanism of hydrogen generation using a Si-based agent. Before the reaction, the Si-based agent was covered with a silicon oxide layer 1–2 nm thick, and this thickness increased as the hydrogen generation reaction proceeded (cf. Reaction (3)) [24]. In the first reaction step, OH^-^ ions move inward through the silicon oxide layer. The inward movement of OH^-^ ions is enhanced by the electrical field induced by the OH^-^ ions adsorbed on the surface of the silicon oxide layer [20]. In the second step, interface Reaction (3) proceeds when OH^-^ ions reach the Si-based agent/silicon oxide interface. The interfacial reactions involve four elementary reactions [25].
2Si + OH^−^ → Si_2_O + H + e(4)
Si_2_O + OH^−^ → 2SiO + H +e(5)
2SiO + OH^−^ → Si_2_O_3_ + H + e(6)
Si_2_O_3_ + OH^−^ → SiO_2_ + H + e(7)

Electrons generated by Reactions (4)–(7) move outward and are trapped in the surface states (i.e., the characteristic energy levels in the bandgap at the surface). Then, water molecules accept these electrons, resulting in the formation of OH^−^ ions and hydrogen. Thus, hydrogen is generated in the interfacial region and at the surface. We observed that hydrogen atoms were bound to the surface, and interfacial Si atoms were present after the hydrogen generation reaction. It is highly probable that these hydrogen atoms are the source of the high reducing power of the Si-based agent. In animal experiments, we observed that the concentrations of reactive sulfur species increased with the administration of the Si-based agent, and it is thought that the hydrogen atoms on Si-based agents play an important role in their formation.

## 3. Antioxidant Effects of the Si-Based Agent

### 3.1. Antioxidant Action via Hydrogen

As described in the previous section, Si-based agents react with water to generate hydrogen [20]. Hydrogen is an antioxidant that specifically scavenges harmful ROS such as hydroxyl radicals. It has become clear that antioxidant therapy using hydrogen (i.e., hydrogen medicine) alleviates the pathology of many diseases, such as brain ischemia–reperfusion (IR) injury, Parkinson’s disease, and hepatitis [17,18]. Since no side effects have been reported thus far [19], hydrogen is expected to be an excellent antioxidant therapeutic agent. The antioxidant action of our Si-based agent resulted from hydrogen generated on the surface of the agent. Because the hydrogen generation reaction of the Si-based agent occurs under neutral to alkaline conditions, activity at the hydrogen generation site is influenced by the pH of the intestinal tract. The method we devised to measure mouse intestinal pH [26] demonstrated that the ileum (pH 7.3) to the rectum (pH 7.8) were somewhat alkaline, and the cecum was the most alkaline at pH 8.4 [21]. Surprisingly, the Si-based agent made the intestinal tract slightly more alkaline than in the normal state, facilitating the generation of hydrogen. The oral administration of the Si-based agent produced hydrogen in the mouse intestinal tract [21]. Moreover, the amount of hydrogen increased in the stomach, large intestine, and small intestine. Of course, the stomach is acidic (pH 3.7), and the results of in vitro experiments show that the Si-based agent does not generate hydrogen under acidic conditions; therefore, it is not conceivable that hydrogen is generated in the stomach. Because of its excellent permeability, hydrogen is thought to easily diffuse out of the intestinal tract from the site of generation [27]. However, because the amount of hydrogen generated by the Si-based agent may have exceeded the amount diffused outside the body, part of the generated hydrogen may have reached the stomach through the intestinal tract. Moreover, the Si-based agent generated a large amount of hydrogen in vitro [20]; however, the rate of increase in hydrogen generation after the Si-based agent was administered into the mouse intestinal tract is not very high [21]. As the capacity of the intestinal tract is fixed, it is conceivable that hydrogen is released outside the body owing to its permeability. In other words, the intestinal tract may have been saturated with hydrogen in the Si-based-agent-administered mice. Not only is it possible to respond to various pathological conditions by continuously transporting antioxidants from the stomach to the large intestine, but this process is also involved in the regulation of autonomic nerves such as hydrogen [28].

It has been reported that both VC and VE are depleted in patients with UC [29,30], and endogenous hydrogen is depleted in UC mouse models [21]. These findings suggest that the onset of UC causes excessive oxidative stress and that hydrogen is consumed and depleted as an antioxidant to relieve oxidative stress. The depletion of hydrogen in the livers of carbon-tetrachloride-induced hepatitis mouse models indicates that hydrogen, like other antioxidants, is used as an antioxidant when oxidative stress occurs [21]. Oxidative stress in the affected area is increased, which requires the elimination of many antioxidants. The administration of a Si-based agent alleviates the aggravation of disease conditions by replenishing the depleted hydrogen in affected areas, such as the large intestine and liver. Hence, Si-based agents are expected to constitute excellent antioxidant treatments for such conditions.

Lipid peroxides also oxidize other substances, and polyunsaturated fatty acids in particular cause ferroptosis, which is iron-dependent cell death [31]. The suppression of the generation and increase in lipid peroxides is necessary to control oxidative stress and cell death. According to previous reports, hydrogen tends to accumulate in the lipid bilayer of cell membranes [32]. Hydrogen is more abundant and soluble in unsaturated fatty acids (such as octanoic acid and linolenic acid) than in water. Calcium influx is inhibited in cells with cell membranes containing dissolved hydrogen. As calcium is a second messenger in cell signal transduction [33], it is possible that hydrogen regulates biological functions and exerts antioxidant effects through calcium signaling.

Because lipid peroxides are among the targets of hydrogen, Si-based agents may also play a major role in mitigating lipid oxidation. Si-based agents alleviate lipid peroxide accumulation in various diseases. The oxidative stress analysis based on lipid oxidation in blood showed that a Si-based agent alleviated systemic oxidative stress associated with pathological conditions in mouse models of UC, chronic renal failure, facial paralysis, and interstitial pneumonia [21,22,23,34]. In particular, we investigated the inhibition of lipid peroxide production in UC mouse models. Si-based agents suppress the increase in early products (hexanoyl-lysine) and lipid peroxides in the blood and increase 4-Hydroxy-2-nonenal (an index of the lipid peroxide production chain reaction) in the colon [21]. Moreover, Si-based agents inhibit the increase in malondialdehyde, the final product of lipid peroxidation, in chronic renal failure, renal ischemia–reperfusion (IR) injury, and IR injury during flap transplantation. Thus, it has been demonstrated that Si-based agents act at various stages of lipid oxidation.

Since Si-based agents inhibit the increase in urinary 8-hydroxy-2-deoxyguanosine in renal disease [22,35,36,37], these results indicate that Si-based agents eliminate various oxidative metabolites, especially lipid peroxides.

### 3.2. Activation of In Vivo Antioxidant Mechanisms

Si-based agents also affect the Nrf2-Keap1 system, an in vivo antioxidant mechanism [38]. Under non-oxidative stress, Nrf2 remains in the cytoplasm, where it binds with Keap1 and undergoes proteasome-dependent proteolytic repression. When cells are exposed to oxidative stress or electrophiles, the binding between Nrf2 and Keap1 is disrupted and Nrf2 is translocated to the nucleus, where it induces the expression of antioxidant genes such as Hmo-1 and cystathionine γ-lyase (Cth) [38]. Increased oxidative stress and decreased endogenous antioxidant factors were observed in the kidneys of chronic renal failure mouse models and the placentas of LPS-induced mother-to-child-infected mice; however, a Si-based agent suppressed the decrease in these factors and alleviated oxidative stress [35,39]. SIRT1, a histone deacetylase, activates Nrf2 [40]. Although SIRT1 expression was decreased in renal failure mouse models, this decrease was mitigated by the administration of a Si-based agent [35]. These findings suggest that Si-based agents directly activate the Nrf2-Keap1 antioxidant system.

Sulfur metabolites (e.g., hydrogen sulfide and glutathione) significantly affect the redox state of living organisms [41]. According to the sulfur index analysis, which exhaustively analyzes sulfur-metabolizing compounds, the large intestines of UC mouse models were oxidized, whereas when the same mice were administered a Si-based agent, their large intestines were in a reduced state. Thus, Si-based agents alleviate the oxidation associated with UC, according to redox indices based on sulfur metabolites [21]. This suggests that Si-based agents have a positive effect on the in vivo redox system involving sulfur metabolites. In particular, an increase in strongly reactive sulfur species (RSS) such as glutathione polysulfide was involved in the suppression of inflammation-associated oxidation in the large intestine in a UC mouse models treated with anSi-based agent. RSS, represented by cysteine persulfide—a molecule with an excess sulfur atom added to the thiol (SH) group of cysteine—is a bioactive substance with strong antioxidant and anti-inflammatory effects and the ability to regulate redox signals [42,43,44,45,46]. In particular, Si-based agents increase the levels of glutathione polysulfides. Glutathione and glutathione peroxidase degrade lipid peroxides into harmless lipid alcohols [47,48]. Therefore, Si-based agents can detoxify lipid peroxides through the glutathione/glutathione peroxidase system.

In infected mouse placenta, a Si-based agent also increased the expression of Cth, a cysteine synthesis enzyme, suggesting the possibility of activating not only RSS but also the entire sulfur metabolism pathway [21,39]. Furthermore, RNA sequencing of kidneys with IR injury revealed that the expression of oxidative-stress-related factors was decreased after the administration of a Si-based agent [37]. In fact, Si-based agents inhibit the decrease in levels of catalase associated with IR, an antioxidant enzyme that neutralizes hydrogen peroxide [49]. Taken together, these results suggest that Si-based agents affect NRF2-Keap1 and the glutathione/glutathione peroxidase systems involving RSS, activating the in vivo antioxidant mechanism.

In conclusion, Si-based agents eliminate oxidative stress by activating the antioxidant action of hydrogen and antioxidant mechanisms in the body, thereby reducing symptoms (Table 1).

## 4. Nano-Antioxidant Therapy

Antioxidant therapy is an effective strategy for treating diseases associated with oxidative stress. However, their efficacy is influenced by how quickly the required amount of depleted antioxidants can be replenished. Furthermore, appropriate antioxidants that specifically neutralize harmful ROS must be developed. It is thought that nano-antioxidant therapy can address these challenges. Currently, methods using platinum nanoparticles [51], GSH-transporting nanoliposomes [52], and high-molecular-weight polymers [53] exist. These methods are useful for enhancing the capacity and specificity of antioxidants. Si-based agents can also be considered nanodevices that carry hydrogen, which is a highly specific antioxidant. Si-based agents with nanoscale particles have been proven to be more effective than those composed of larger particles [37]. This is because water reacts with the silicon on the surface of the agent to generate hydrogen [22], thus reducing the size of the Si-based agent, increasing the surface area, and generating more hydrogen. Consequently, their effectiveness has increased in recent years. Thus, we believe that nano-antioxidant therapy is an innovative approach that can overcome the problems limiting existing antioxidant therapies. Nanotherapy is also used as an anti-cancer treatment and has the potential to significantly improve the treatment of many diseases. In the future, Si-based agents will contribute to the development of nanotherapeutic methods.

## 5. Other Actions of Si-Based Agents

### 5.1. Anti-Inflammatory Effects

It has been determined that Si-based agents also have anti-inflammatory effects in addition to their antioxidant effects. Si-based agents suppress the increase in proinflammatory cytokines (Interleukin (IL)-6, IL-1β, tumor necrosis factor-α (TNF-α), Interferon (IFN)-γ, and IFN-α), chemokines (C-C motif chemokine ligand 2: CCL2), and proinflammatory factors (inducible nitric oxide synthase: iNOS) observed during inflammation. In particular, the increased expression of IL-6 associated with pathological conditions is suppressed by Si-based agents in many diseases associated with inflammation, such as renal failure, renal IR injury, IR injury during flap transplantation, UC, mother-to-child infections, and neuropathic pain [21,22,37,39,50,54,55]. As IL-6 promotes acute-phase inflammation in various organs, the suppression of IL-6 expression is useful in mitigating pathological conditions [56]. Moreover, RNA sequencing of the kidneys of renal IR injury rat models showed that treatment with a Si-based agent tended to suppress both immune responses and signaling pathways for cytokine production. In fact, Si-based agent administration inhibited the increase in the proinflammatory factors (Intercellular adhesion molecule 1: ICAM1; Phorbol-12-Myristate-13-Acetate-Induced Protein 1: PMAIP1) and the decrease in Peroxisome Proliferator-Activated Receptor α (PPARα), an anti-inflammatory factor [37]. Thus, by regulating the changes in inflammation-related factors caused by pathological conditions, Si-based agents tend to alleviate inflammation. The infiltration of macrophages and neutrophils into the affected colon, placenta, and lung was suppressed in UC, mother-to-child infection, neuropathic pain and intestinal pneumonitis mouse models, respectively, resulting in the alleviation of colon atrophy, a reduced incidence of miscarriage, the amelioration of the trigeminal ganglion demyelination and the inhibition of pulmonary interstitial fibrosis [21,34,39,55]. In particular, in rats with neuropathic pain, Si-based agents inhibited the activation of the inflammatory Capsae-1 via the NLR family and the pyrin domain-containing 3 protein (NLRP3) inflammasome, thereby suppressing microglia pyroptosis, which leads to the activation of inflammation [55]. Interestingly, imaging analyses, such as Magnetic Resonance Imaging and Magnetic Resonance Angiography, revealed that inflammation of the colon in UC and the corresponding associated vasodilation were suppressed [21]. It is of considerable value that we succeeded in capturing the Si-based-agent-mediated suppression of inflammation in living mice in real time. In another study, inflammation was induced by poly (I:C) in the brains of mice during mother-to-child infection research; however, oxidative stress did not occur [54]. Taken together, these results demonstrate that the Si-based agents directly suppress inflammation (Table 2).

What is responsible for the anti-inflammatory effects of Si-based agents? One hypothesis is that these effects are hydrogen-mediated. Hydrogen suppresses inflammation by suppressing proinflammatory cytokines [17,18]. Conversely, it has been found that RSS also negatively regulate pro-inflammatory effects mediated by Toll-like receptors and thereby protect cells from inflammation [57]. Both of these factors likely contribute to the observed anti-inflammatory and antioxidative effects. Taken together, these results indicate that Si-based agents exhibit not only antioxidant effects but also anti-inflammatory effects.

### 5.2. Anti-Apoptotic Effects

The apoptosis-induced loss of constituent cells leads to organ dysfunction, resulting in the onset and aggravation of diseases. Therefore, the suppression of apoptosis is very effective for disease treatment. In rat models of renal failure, the administration of an Si-based agent suppressed caspase 3, the final apoptosis-execution factor [22,34]. In rat models of kidney IR injury, the downregulation of the apoptotic signaling system was observed in the Si-based-agent-treated group compared to the non-treated group [37]. In fact, the Si-based agent alleviated renal IR injury, cell death in renal failure, and the fragmentation of sperm DNA in testicular varicose veins [22,35,36,37]. Moreover, a Si-based agent suppressed cell death caused by IR injury during flap transplantation, resulting in a reduced flap necrosis area and the alleviation of flap injury [50]. Furthermore, in mouse models of Parkinson’s disease, a Si-based agent inhibited the 6-hydroxydopamine-induced cell death of nigral dopamine neurons and alleviated motor coordination deterioration [22]. Interestingly, the expression of Pstg2, an important anti-apoptotic factor in the placenta, was increased by the administration of a Si-based agent and was significantly involved in alleviating placental structural collapse caused by mother-to-child infection [39]. In particular, an increase in Pstg2 expression above the normal state was observed after the administration of a Si-based agent.

Taken together, it is clear that Si-based agents exhibit anti-apoptotic effects, in addition their antioxidant activity (Table 3).

### 5.3. Anti-fibrotic Effects

Renal fibrosis observed in chronic renal failure is a tissue change mainly involving renal tubular atrophy and interstitial fibrosis, which promote progression to end-stage renal failure [58]. Therefore, reducing renal fibrosis is a promising therapeutic strategy for the treatment of chronic renal failure. A Si-based agent suppressed the increased expression of profibrotic factors such as tissue inhibitor of metalloproteinase-1 (TIMP1) and α-smooth muscle actin (α-SMA) in renal failure rat models. Although renal fibrosis reduces renal function, a Si-based agent alleviated glomerular hypertrophy and proteinuria and caused the increase in blood creatinine observed during functional decline, which was associated with the progression of renal failure [22,35]. In rat models of renal failure, IL-6 expression was increased, and fibrosis was promoted through increased collagen expression [22,35]; however, it was suggested that a Si-based agent may alleviate fibrosis by suppressing its expression [22]. Moreover, in interstitial pneumonia mouse models, a Si-based agent significantly alleviated pulmonary fibrosis by suppressing the increased expression of α-SMA associated with the pathology [34]. Collectively, these results suggest that Si-based agents suppress interstitial fibrosis (Table 3).

The main action of Si-based agents is their antioxidant effect, but the findings obtained thus far in various studies using disease mouse models have suggested that Si-based agents may have other actions. Indeed, during inflammation caused by an immune reaction, oxidative stress is enhanced by ROS released from immune cells; therefore, inflammation and oxidative stress are closely related [59]. Consequently, a reduction in oxidative stress leads to the suppression of inflammatory effects. Moreover, as the damage caused by oxidative stress induces apoptosis, alleviating oxidative stress alone suppresses apoptosis [60]. However, the anti-inflammatory effect observed in mother-to-child infection mouse models, which was not directly related to oxidative stress—as well as the high expression of the anti-apoptotic factor Ptgs2 in the placenta in the normal state—suggests that Si-based agents have direct anti-inflammatory and anti-apoptotic effects. In contrast, macrophage infiltration and increased oxidative stress related to chronic inflammation are significantly associated with fibrosis. Therefore, the anti-fibrotic effects of Si-based agents may be the result of their antioxidant and anti-inflammatory effects [61]. In any case, because the strong suppression of fibrosis via two types of action—antioxidation and anti-inflammation—leads to the mitigation of pathological conditions, Si-based agents have great potential as a therapy for fibrosis.

## 6. Hypothetical Mechanism of the Spreading of Si-Based Agent Effects into Various Organs

Our Si-based agent has effects in various organs such as the kidneys, skin, placenta, and uterus, in addition to the large intestine, which is the site of hydrogen generation (Table 4). Surprisingly, it also affects the brain, which is distal to the large intestine. We found that it ameliorated central nervous system abnormalities, for example, by alleviating dopamine neuron loss and decreased motor coordination in Parkinson’s disease mouse models; suppressing nausea and vomiting in cisplatin-treated mice; and alleviating visceral pain and discomfort in UC mouse models [21,22,62]. Moreover, the Si-based agent also exhibited nerve repair effects, including the promotion of myelin formation and the recovery of facial nerve function following facial nerve injury [23].

To date, it remains unclear whether the antioxidant effects of Si-based agents directly affect the brain. Indeed, it is possible that the effects of Si-based agents are exerted on organs other than the large intestine via blood circulation. Hydrogen has also been shown to be effective in treating pathological conditions in various organs, including the brain [17,18]. Since the main methods of in vivo hydrogen administration involve either inhaling hydrogen gas or drinking hydrogen-rich water, it is conceivable that hydrogen acts indirectly on organs other than the lungs and digestive tract. According to previous reports, the detection of hydrogen in the blood [64] and the possibility that heme protein is a direct target of hydrogen [65] also suggest that hydrogen exerts antioxidant effects through the blood. In fact, previous studies have confirmed the presence of hydrogen in the blood when a Si-based agent is administered. Moreover, the blood hydrogen concentration increases in proportion to the administered quantity of the Si-based agent, supporting the idea that it acts on the brain via blood circulation [35,37].

However, Si-based agents also increase RSS. It has been reported that RSS are produced in vivo by intestinal bacteria and subsequently enter the bloodstream [66]. Therefore, it is conceivable that not only hydrogen but also RSS can reach other organs via the blood. There are still many issues to be investigated, such as the effects of Si-based agents on the intestinal flora and the autonomic nervous system. However, in view of our findings so far, it is highly likely that the effects of Si-based agents, including their antioxidant effects, are exerted on organs other than the intestinal tract via the blood (Figure 4).

## 7. Expectations for Clinical Applications of Si-Based Agents

Because there have been no reported adverse effects of hydrogen [19], it is expected that Si-based agents, too, will not cause side effects. In fact, various safety tests (i.e., a 91-day repeated-dose toxicity test in rats, methylthiazole tetrazolium assay, chromosomal aberration test, and reverse mutation test) have shown that Si-based agents do not exhibit any side effects or toxicity [22]. Because Si-based agents are powders that generate hydrogen only when they react with water, they are easier to store than hydrogen water or hydrogen gas, and there is no risk of explosion. Therefore, they can be safely administered at home. Because hydrogen readily permeates many materials, a large amount of hydrogen continuously generated by a Si-based agent is released from the body’s surface. Therefore, unlike other antioxidants, Si-based agents do not cause side effects in the body at high doses [11,12,13].

The amount of hydrogen in the blood is proportional to the concentration of the Si-based agent, and the therapeutic efficacy for pathological conditions also increases with concentration [22,35]. Efficacy evaluations of Si-based agents using UC mouse models showed efficacy at 0.025% content in the diet, and increasing the concentration showed efficacy against UC with severe inflammation [21]. Si-based agents exhibited effectiveness at different doses depending on the disease condition. They offer a solution to the problem in hydrogen medicine of how to fill the body with hydrogen for extended periods. Moreover, because Si-based agents are effective against diseases of various organs, they can exert therapeutic effects on areas other than the principally affected areas for a given condition, such as neurological symptoms associated with UC [21]. Taken together, it is expected that the antioxidant and anti-inflammatory effects of Si-based agents will render them epoch-making therapeutic agents that exert their effects not only at the injury site, but also distally, thereby systemically ameliorating pathological conditions.

Si-based agents have also been shown to not interfere with the effectiveness of other drugs. Methylcobalamin (active vitamin B12) is often used in the treatment of facial nerve paralysis; however, the combined administration of a Si-based agent was found to be more effective than the administration of either agent alone [23]. Surprisingly, both drugs exhibited antioxidant and anti-inflammatory effects but did not inhibit each other. Steroids are often administered for the treatment of facial nerve paralysis, but they are difficult to use in the elderly because of their severe side effects. Combination therapy may be a solution to these problems.

Elsewhere, Si-based agents were also effective against nausea and vomiting, which is a major side effect of anti-cancer drugs [62]. As hydrogen does not inhibit the action of anti-cancer drugs [67], it is predicted that it will be possible to administer Si-based agents together with anti-cancer drugs. Since nausea and vomiting associated with anti-cancer drug treatment can be very distressing in patients undergoing chemotherapy, the administration of anti-emetic drugs is important. Currently, serotonin type 3 receptor antagonists are often used as vomiting control agents; however, serotonin itself is very important for the nervous system, and such drugs may affect other neural functions. Therefore, the use of Si-based agents as new anti-emetic agents is expected to improve the quality of life in many chemotherapy-treated patients.

## 8. Limitations

Because Si-based agents generate hydrogen from their surfaces, it is extremely difficult to label them with fluorochromes or radioactive elements and examine their in vivo kinetics, resulting in decreased activity via the inhibition of surface treatment. Moreover, hydrogen generated from Si-based agents and ROS, such as hydroxyl radicals, is colorless, odorless, and exists for only a short time; therefore, it cannot be visualized, making it difficult to directly elucidate the mechanism of action.

Moreover, since Si-based agents differ from general drug candidates in terms of their mechanism of efficacy, there are many hurdles to overcome for drug development. Si-based agents generate hydrogen in the intestine, but the agent itself is not absorbed and then excreted as stool. Therefore, they are difficult to evaluate via pharmacokinetic studies in non-clinical tests.

Nevertheless, further research to characterize factors that bring efficacy to the brain, new efficacy, safety profiles, etc., remains a priority.

## 9. Conclusions

Si-based agents, produced through the interdisciplinary research, are new antioxidant substances formed from inorganic precursors. They also constitute an epoch-making bioadministration method for the highly effective and selective antioxidant hydrogen, as they can sustainably generate large amounts of hydrogen in the intestinal tract after oral administration. Si-based agents also activate in vivo antioxidants, including RSS, and their effectiveness has been demonstrated in animal models of various pathologies, such as UC, renal failure, Parkinson’s disease, facial paralysis, and mother-to-child infections.

Moreover, Si-based agents exert a range of anti-inflammatory, anti-apoptotic, and anti-fibrosis effects. Importantly, hydrogen has no reported side effects, rendering it an excellent antioxidant therapeutic drug candidate for use by men and women of all ages. Furthermore, RSS derivatives that can increase RSS production are attracting attention as next-generation drug discovery targets, and Si-based agents are expected to be promising candidates in this connection. Additionally, they are attractive because they can be safely administered in combination with other drugs.

However, there are some issues in developing Si-based agents as pharmaceuticals. The first is the identification of therapeutic target factors on which the hydrogen generated from the agent acts directly. The second is the significance of hydrogen newly generated by the Si-based agent in vivo. Originally, hydrogen-producing bacteria, which are intestinal bacteria, produce hydrogen in vivo, so a certain amount of hydrogen is detected in the intestinal tract [21,68]. The amount of hydrogen in the large intestine after the administration of a Si-based agent was about 1.3 times that of an unadministered one. It is very mysterious that this small amount of increase in hydrogen in the intestinal tract shows great efficacy against various diseases. The third is to elucidate the relationship between Si-based agents and increased RSS. RSS derivative candidates so far contain sulfur, which is the material of RSS, but Si-based agents do not contain sulfur. It remains unclear how RSS, a potent endogenous antioxidant, is induced by Si-based agents.

Solving these problems will significantly lead to the elucidation of the action mechanism of Si-based agents. If the action mechanism of Si-based agents can be clarified, the development of SI-based agents as pharmaceuticals will be greatly advanced. Therapeutic methods using novel antioxidant Si-based agents may become new treatment options for various diseases in the future.

## 10. Future Directions

Si-based agents are still in the research stage using animal disease models, and the only clinical specification to date is a report on the improvement of the intestinal environment of cancer patients following the administration of a Si-based agent [63]. Therefore, the efficacy of Si-based agents should be investigated further in future clinical studies. However, a lot of knowledge acquired from disease animal models may greatly contribute to the improvement of existing treatment methods and the development of new treatment methods. In fact, silicon nanoparticles for oral administration have been reported, referring to the findings of Si-based agents [69]. Moreover, there have recently been reports of medical application papers using silicon nanoparticles [70]. We hope that this review will stimulate further revitalization of this research field and that the day will come when the action mechanism of Si-based agents will be elucidated.

## Figures and Tables

**Figure 1 antioxidants-12-01061-f001:**
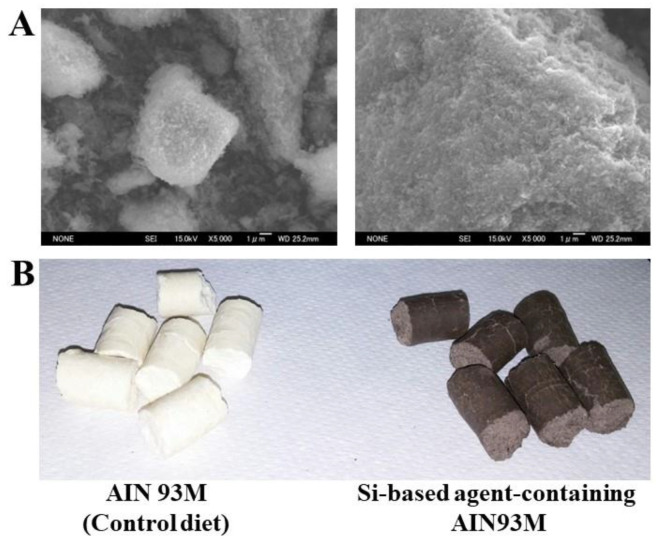
(**A**) Scanning electron micrographs of two different specimen positions of Si-based agent consisting of agglomerates of Si nanopowder which was fabricated via the bead-milling method and surface treatment using a hydrogen peroxide oxide solution. Small (**left**) and large (**right**) agglomerates. (**B**) Specially ordered diets fed to the experimental animals. AIN93M (**left**) and AIN93M containing 2.5% Si-based agent (**right**).

**Figure 2 antioxidants-12-01061-f002:**
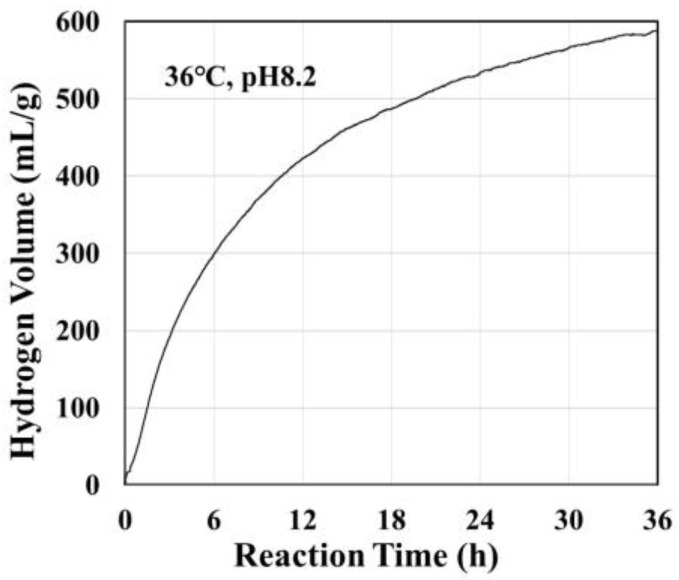
Volume of hydrogen generated by the reaction of Si-based agent with a pH 8.2 solution at 36 °C vs. the reaction time. The volume of generated hydrogen was determined by measurements of the dissolving hydrogen concentration using a potable dissolving hydrogen-meter. The solution was confined in a stainless container in which no gas phase was present, and therefore, all generated hydrogen dissolved in the solution.

**Figure 3 antioxidants-12-01061-f003:**
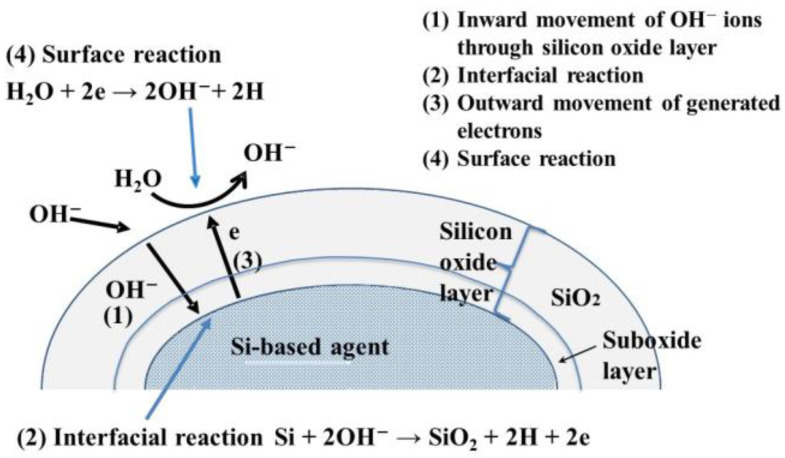
Mechanism of hydrogen generation by the reaction of Si-based agent with aqueous solutions, e.g., pancreatic juice and intestinal juice.

**Figure 4 antioxidants-12-01061-f004:**
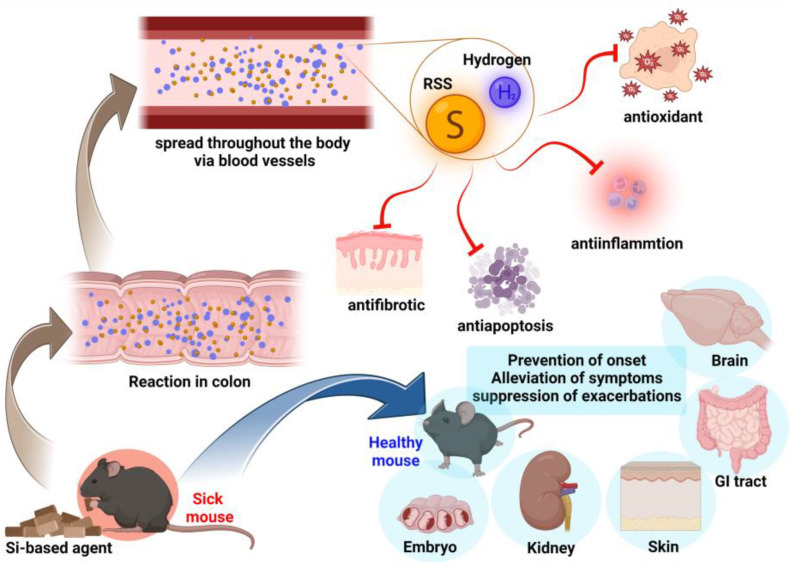
Hypothesis of Si-based agents’ mechanism of action. Si-based agents orally administered to mice via diet react with intestinal juice in the large intestine, resulting in the generation of large amounts of hydrogen and the promotion of RSS production. Hydrogen and RSS are spread throughout the body via the blood. They exert antioxidant, anti-inflammatory, anti-apoptotic, and anti-fibrotic effects on diseased organs (brain, kidneys, intestinal tract, etc.) to improve pathological conditions. RSS: reactive sulfur species. GI tract: gastrointestinal tract. Created using BioRender.com.

**Table 1 antioxidants-12-01061-t001:** Antioxidant effects of Si-based agents against diseases.

	DISEASE	Administration of Si-Based Agent	Refs. No.
**Pro-oxidant**	**The increase in oxidative stress in blood**	**Alleviation**	**[21,22,23,34]**
	**The increase in LPO (HEL, 4-HNE, MDA)**	**Suppression**	**[21,35,37,50]**
	**The increase in urinary 8-OHdG**	**Inhibition**	**[22,35,36,37]**
	**The oxidation in large intestine**	**Alleviation**	**[21]**
**Antioxidant**	**The decrease in Nrf2-Keap1 system-induced factors (Hmox-1, Cystationine-γ-lyase)**	**Inhibition**	**[35,39]**
	**The decrease in RSS ***	**Alleviation**	**[21]**
	**The decrease in SIRT1**	**Alleviation**	**[35]**
	**The decrease in Catalase**	**Alleviation**	**[37]**

* The increase via the administration of Si-based agent even in normal conditions.

**Table 2 antioxidants-12-01061-t002:** Anti-inflammation effects of Si-based agents against diseases.

	DISEASE	Administration of Si-Based Agent	Refs. No.
**Proinflammation**	**The increase in proinflammatory cytokines (IL-6, IL-1β, TNF-α, IFN-γ, IFN-α, CCL-2)**	**Inhibition**	**[21,22,37,39,50,54,55]**
	**The increase in ICAM1, iNOS**	**Inhibition**	**[37]**
	**Activation of NLRP3 inflammasome**	**Inhibition**	**[55]**
	**Activation of Proinflammatory Caspase1**	**Inhibition**	**[55]**
	**Immunocyte invasion**	**Inhibition**	**[21,34,39,55]**
	**Vasodilation (MRA)**	**Alleviation**	**[21]**
**Antii-inflammation**	**The decrease in PPARα**	**Alleviation**	**[37]**
	**The decrease in RSS**	**Alleviation**	**[21]**

**Table 3 antioxidants-12-01061-t003:** Anti-apoptosis and anti-fibrotic effects of Si-based agents against diseases.

	DISEASE	Administration of Si-Based Agent	Refs. No.
**Pro-apoptosis**	**The increase in Caspase 3**	**Inhibition**	**[22,35]**
	**Apoptosis** **(TUNEL-positive, DNA fragmentation)**	**Inhibition**	**[22,36,37,50]**
	**The increase in Pmaip1**	**Inhibition**	**[37]**
**Anti-apoptosis**	**No change in Ptgs2**	**Increase ***	**[54]**
**Profibrotic**	**The increase in α-SMA**	**Inhibition**	**[22,34,35]**
	**The increase in TIMP1**	**Inhibition**	**[22,37]**

* The increase via the administration of Si-based agent even in normal conditions.

**Table 4 antioxidants-12-01061-t004:** List of publications demonstrating effectiveness of Si-based agents against diseases.

PAPER	EXPERIMENTAL OBJECT	Anti Oxidant	Anti-Inflammation	Anti-Apoptosis	Anti Fibrotic	RESULTS	Ref. No
**Kobayasi Y et al.,** **Sci Rep. 2020**	**Renal failure rat** **remnant kidney model**	**〇**	**〇**	**〇**	**〇**	**Alleviation of renal fibrosis** **and renal dysfunction**	**[22]**
	**Parkinson’s disease** **mouse model**	**-**	**-**	**〇**	**-**	**Alleviation of decreased** **motor coordination**	
**Imamura R et al.,** **Biochem Biophys Res** **Commun. 2020.**	**Renal failure rat** **remnant kidney model**	**〇**	**〇**	**〇**	**〇**	**Alleviation of renal fibrosis** **and renal dysfunction**	**[35]**
**Kawamura M et al.,** **Front Med** **(Lausanne). 2020.**	**Renal IR injury** **rat model**	**〇**	**〇**	**-**	**-**	**Alleviation of** **renal dysfunction**	**[37]**
**Inagaki Y et al.,** **Andrology. 2021**	**Varicocele** **rat model**	**〇**	**-**	**-**	**-**	**Improvement of sperm** **motility and fertility**	**[36]**
**Usui N et al., Front** **Med Technol. 2021.**	**Maternal immune** **activation mouse model**	**〇**	**〇**	**-**	**-**	**Alleviation of miscarriage miscarriage rate**	**[39]**
**Tanaka Y et al.,** **nutrients. 2021.**	**Cancer patient**	**-**	**-**	**-**	**-**	**Alleviation of** **renal dysfunction**	**[63]**
**Usui N et al.,** **Front Psychiatry. 2022.**	**Maternal immune** **activation mouse model**	**-**	**〇**	**〇**	**-**	**Alleviation of social decline in newborns**	**[54]**
**Otani N et al.,** **Sci Rep. 2022.**	**Skin-flap IR injury rat model**	**〇**	**〇**	**〇**	**-**	**Improvement of the survival** **rate of grafted skin flaps**	**[50]**
**Koyama Y et al.,** **Sci Rep. 2022.**	**UC mouse model**	**〇**	**〇**	**-**	**-**	**Relief of** **colitis symptoms**	**[21]**
**Yanagawa H et al.,** **Biochem Biophys** **Rep. 2022.**	**Cisplatin-induced** **nausea & vomiting** **mouse model**	**-**	**-**	**-**	**-**	**Relief of** **nausea and vomiting** **with anticancer drugs**	**[62]**
**Koyama Y et al.,** **Biochem Biophys** **Rep. 2022.**	**Facial paralysis** **mouse model**	**〇**	**-**	**-**	**-**	**Promotes recovery** **of nerve function**	**[23]**
**Guo Mu et al., Int Immunopharmacol. 2023.**	**Neuropathic pain** **rat model**	**-**	**〇**	**-**	**-**	**Amelioration of** **neuropathic pain**	**[55]**
**Shimada M et al.,** **Sci Rep. 2023.**	**Interstitial pneumonitis** **mouse model**	**〇**	**〇**	**-**	**〇**	**Relief of** **pulmonary fibrogenesis**	**[34]**

**〇: effective; -: uninvestigated.**

## Data Availability

Not applicable. All relevant data are within the paper.

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
