# Peer review of "Diverse Possibilities of Si-Based Agent, a Unique New Antioxidant"

_antioxidants, 2023, doi:10.3390/antiox12051061_

Round 1
Reviewer 1 Report
This is an interesting review devoted to a fascinating product, written in a very pedagogical manner.
Nevertheless, I have two major remarks and a minor suggestion:
1) as it is, it sometimes sound more like an advertising than a fair assessment of the literature in this field. Without underestimating the pioneer work and key position of the authors in this field, I would guess that they are not the only ones to have developed and studied silicon particles for the same application. Some presentation/comment of others' works and results is necessary to provide a full picture of this area.
2) At the end of this review, after reading so much about the many properties of this product, I asked myself about its clinical application. Did it receive any agreement by regulation agencies ? If not, what are the current challenges/limitations towards commercialization ? I think this should be made clear, all the more as some of the authors received funding from a big pharmaceutical company
3) as a chemist, I would really appreciate to have a bit more details about the used particles (size, morphology,...). The authors provide cristalline size but, if it is in the crystallographic sense, then it does not give any indication about the particles. I guess some microscopy (SEM ?) picture would be much more informative than Figure 1.
Author Response
Thank you for peer reviewing. Please see the attachment.

Reviewer 2 Report
The manuscript is a review of, what seems to be, the papers of the same authors or the same research group. The impression I got is that the authors try to promote and cite their own articles by this review. Table 1 lists eleven publications involving the Si-based agent. Ten of these eleven publications are written by the authors of the review. It is interesting to note, that one publication [65] by other researchers has nothing to do with the Si-based agent.
My advice would be against publishing such type of review where authors are clearly focused on their own publications.
If the manuscript gets accepted, the authors should check the singular – plural grammar related to „Si-based agent“. It seems that there is just one Si-based agent, but in many places in the manuscript plural is used in association with this agent.
Author Response

(The authors gave the same response as above.)

Reviewer 3 Report
The current review paper entitled Diverse Possibilities of Si-based Agent, a Unique New Anti-oxidant fits within the general scope of the Journal. It covers an important topic about the importance of silicon-based agents. The review paper was well organized and written and deserves publication in Antioxidants MDPI after taking into consideration the following minor comments and suggestions:
1) the abstract section is not well written and is not informative since the authors should be indicated what is the main topic that they have covered and a final sentence should be inserted in the revised version of the manuscript as a take-home message and the importance of this review paper for the scientific community.
2) The authors at the end of the Introduction should clearly state the hypothesis in other words what is the novel part of this review compared to the previously published papers.
3) I urge the authors to summarize the results of several sections of this review paper into tables which will be easier and clearer for the readers of Antioxidants.
4) The limitations section is very short and should be expanded.
5) The conclusion section should be also expanded and the authors should clearly report the challenges ahead and not repeat what was written in the previous section.
Good luck!
Author Response

(The authors gave the same response as above.)

Round 2
Reviewer 1 Report
The authors did significant efforts to improve their manuscript.
Although I still think the paper lacks a more open presentation/discussion of the use of Si particles in the biomedical field (i.e. not restricted to antioxidant applications), it nevertheless gathers interesting data.
There is just one important thing to correct in the current version, which is related to the new SEM images. The caption indicates that left and right hand images in Figure 1A are at low and high magnification. However, looking at the scale bar and magnification, they were recorded in exactly the same conditions, i.e. we are just looking at small and large grains. Please check and correct
Author Response
Thanks for pointing out the mistakes in scanning electron micrographs. We were unaware of this mistake. As you pointed out, these are micrographs of large or small particles of Si-based agent taken at the same magnification.
We rewrote the legend of Figure 1 (blue marker). We are using a proofing tool of Word, so please check.
Thanks to you, the review has become even better.
Reviewer 2 Report
The authors have revised the manuscript. The inclusion of other researcher's works has elevated the present review into a usual standard for the reviews.
Author Response
Thanks to your advice, the content of the review was very fulfilling.
We sincerely appreciate the peer review.